# Effects of UV Irradiation on the Sensing Properties of In_2_O_3_ for CO Detection at Low Temperature

**DOI:** 10.3390/mi10050338

**Published:** 2019-05-22

**Authors:** Lucio Bonaccorsi, Angela Malara, Andrea Donato, Nicola Donato, Salvatore Gianluca Leonardi, Giovanni Neri

**Affiliations:** 1Dipartimento DICEAM, Università Mediterranea, Loc. Feo di Vito, 89060 Reggio Cal, Italy; angela.malara@unirc.it (A.M.); andrea.donato@unirc.it (A.D.); 2Dipartimento di Ingegneria, Università di Messina, C.da Di Dio, 98166 Messina, Italy; leonardis@unime.it (S.G.L.); gneri@unime.it (G.N.)

**Keywords:** indium oxide, UV irradiation, CO detection

## Abstract

In this study, UV irradiation was used to improve the response of indium oxide (In_2_O_3_) used as a CO sensing material for a resistive sensor operating in a low temperature range, from 25 °C to 150 °C. Different experimental conditions have been compared, varying UV irradiation mode and sensor operating temperature. Results demonstrated that operating the sensor under continuous UV radiation did not improve the response to target gas. The most advantageous condition was obtained when the UV LED irradiated the sensor in regeneration and was turned off during CO detection. In this operating mode, the semiconductor layer showed an apparent “p-type” behavior due to the UV irradiation. Overall, the effect was an improvement of the indium oxide response at 100 °C toward low CO concentrations (from 1 to 10 ppm) that showed higher results than in the dark, which is promising to extend the detection of CO with an In_2_O_3_-based sensor in the sub-ppm range.

## 1. Introduction

Indium oxide is a semiconductor metal oxide (MOX) which displays very good performances when used as a sensing layer in resistive gas sensors for the detection of oxidizing gases like O_3_ and NO_2_ [1,2,3,4,5]. In_2_O_3_ has been investigated less for CO detection at very low concentrations. Further, previous In_2_O_3_-based CO sensors needed to be operated at elevated temperatures, in the range of 250–400 °C [6,7,8,9]. In a previous paper, we reported a fast and repeatable response toward this gas, allowing the detection of less than 2 ppm of CO in a few seconds, operating the sensor at 250 °C [10]. As the working temperature is one of the important parameters that determines the effective use of a sensor in commercial applications, the development of sensors for the monitoring of very low CO concentrations at low temperature is presently an active research field because these devices have lower operating costs, longer lifespans and an increased stability [11,12].

To improve performances of resistive In_2_O_3_ sensors at lower temperatures, different routes are possible. Doping indium oxide with low concentrations of additives has been proved to be beneficial in promoting sensitivity and lowering operating temperatures [2,5,6,7,8]. UV irradiation has been also successfully reported in recent papers for improving the performances of the In_2_O_3_ sensing surface [13,14,15,16,17]. Indium oxide, indeed, is a UV responsive metal oxide since UV photons radiation causes the formation of electron/hole pairs in the depletion region of the oxide grains that increase the intra-grain conductivity of the sensing layer [18]. In detection of oxidizing gases such as NO_2_, it is well known that the UV radiation of the semiconductor oxide surface causes larger variations of the electrical resistance, that in turn improves the sensing performance of In_2_O_3_ at lower temperatures [6,16,19].

However, very little data is available about the effect of UV in promoting CO sensing on In_2_O_3_ and the related phenomena involved [13]. It is known, indeed, that UV illumination has multiple effects on oxygen adsorption, conductance of the sensing layer and CO adsorption/reactivity [13,15,17]. In this work, we report the results of an investigation on CO sensing properties of In_2_O_3_ under different UV irradiation modes to improve MOX response at low temperatures. CO concentrations in the range of 1–10 ppm and sensor temperatures from 25 °C to 150 °C were used in combination with UV light activated during CO detection or not. Results showed that UV light generally did not improve In_2_O_3_ response at low temperatures, however, when used in certain conditions, the UV photons irradiation demonstrated to increase the sensor response. This demonstrates the possibility to detect sub-ppm concentrations of CO with In_2_O_3_-based sensor at a relatively low temperature without any use of additives. 

## 2. Materials and Methods

### 2.1. Sample Preparation and Characterization

Indium oxide powder was prepared by precipitation from an aqueous solution of indium nitrate (0.68 M) hydrolyzed with an aqueous potassium carbonate solution (1 M). The obtained precipitate was filtered, washed with deionized water, dried at 110 °C for 12 h and then calcined at 500 °C for 12 h in air [7].

Powder sample was characterized by X-ray powder diffraction (XRD) analysis (Bruker, D2 Phaser, Karlsruhe, Germany) in the 2θ range from 10 to 80° (Cu Kα1 = 1.54056 Å) and its morphology studied by Scanning Electron Microscopy SEM (Phenom ProX, Deben, Suffolk, UK). The Brunauer–Emmett–Teller (BET) surface areas and the total Pore Volume of the prepared In_2_O_3_ powder were determined from nitrogen adsorption–desorption isotherms at 77 K (ChemiSorb 2750 Micromeritics, Norcross, GA, USA).

### 2.2. Sensor Preparation and Testing

Sensors were prepared by depositing a paste of indium oxide powder mixed with a proper quantity of ethanol onto an alumina planar substrate (3 mm × 6 mm) supplied with interdigitated Pt electrodes and a heating element on the back side. Before sensing tests, the sensors were conditioned in air for 2 h at 400 °C to stabilize the deposited film. Measurements were performed positioning the sensor in a testing cell and flowing a mixture of dry air and CO at different concentrations for a total gas stream of 100 sccm. All gas fluxes were measured by computer-controlled mass flow meters. The sensors resistance data were collected in the four-point mode by an Agilent 34970A (Santa Clara, CA, USA) multimeter while a dual-channel power supplier instrument (Agilent E3632A, Santa Clara, CA, USA) allowed controlling the sensor temperature.

The testing cell was equipped with an UV LED (λ = 400 nm, I = 20 mW/cm^2^) sited in front of the sensing layer of the sensor. The UV irradiation of the sensor surface was varied during the experiments, according to the diagram shown in Figure 1. Three different CO concentrations and sensor temperatures have been tested using four different UV irradiation modes for each case: UV (always) Off, UV (always) On, UV On in sensor regeneration (airflow), UV On in detection (air+CO) (Figure 1).

## 3. Results and Discussion

### 3.1. Morphological and Microstructural Characterization

The morphology of the synthesized In_2_O_3_ powder was investigated by SEM analysis. The image in Figure 2 shows that the In_2_O_3_ powder is constituted by small particles with a characteristic cubic shape, assembled in larger agglomerate with wide size distribution ranging from 0.1 to 0.8 μm. 

XRD analysis demonstrated that these particles are highly crystalline. The XRD spectrum of precipitated powder after calcination (T = 500 °C) confirmed the formation of crystalline indium (III) oxide in the typical cubic phase, bixbyite (Figure 3). The crystallites size calculated by the Scherrer formula from diffraction peaks in Figure 3 resulted ~30 nm, confirming that the larger granules observed by electron microscopy are aggregates.

The above morphological and microstructural characteristics suggest that this sample likely has a large surface area. In fact, the BET surface area of the precipitated indium oxide, measured by nitrogen adsorption method, was 20.3 m^2^/g and the total Pore Volume = 87 mm^3^/g. For comparison, the surface area of a commercial In_2_O_3_ (Sigma Aldrich) resulted much lower: only 10.2 m^2^/g with a pore volume of 42 mm^3^/g.

### 3.2. Gas Sensor Measurement

Indium oxide is a MOX semiconductor showing typical n-type behavior, i.e., the electrical resistance of the deposited layer decreases in presence of CO due to the oxidation of the reducing target gas on the MOX surface. Measurements carried out in dark conditions (see Figure 4 and Figure 5) confirm this behavior.

Figure 4 shows the sensor resistance variation with temperature for a concentration of 5 ppm CO. Although the temperatures range considered was low, the baseline resistance in the airflow was lower than 1 kΩ even at T = 25 °C and decreased increasing the sensor temperature to 100 and 150 °C (Figure 4). The low values shown by In_2_O_3_ are due to the high intrinsic electron concentration with good mobility in the sensing layer of this semiconductor metal oxide [20,21].

The transient response at T = 25 °C and 100 °C was weak but showed a complete and reversible recovery in few minutes at the removal of the target gas (Figure 4). Increasing the temperature to 150 °C, the recovery time to the baseline was similar although the resistance variation increased. Comparable values were observed at the different CO concentrations, as shown by the sensor transient response at T = 100 °C in Figure 5. The recovery time of the In_2_O_3_ layer was not influenced by the CO concentration, showing that the adsorption/desorption and reaction processes occurring on the sensor surface were unaffected. 

To evidence the sensor behavior, the response S = R_0_/R, where R_0_ is the baseline resistance in air and R is the resistance at different CO concentrations, is plotted for the three tested temperatures in Figure 6. In “UV Off” mode (Figure 6a), the response at room temperature was very low, however it increased significantly by heating the sensor surface at 100 °C and even more at 150 °C.

The ideal working temperature for resistive sensors is generally higher than 200 °C [6,7,8,9,10] and considering the low temperatures and the CO concentrations range tested, the In_2_O_3_ sensor has however shown a weak but detectable sensitivity.

In the same conditions, the continuous irradiation of the sensor surface, i.e., in the “UV On” mode (Figure 6b), was never beneficial. Indeed, comparing data in Figure 6, the sensor response in “UV On” was similar or even lower than in dark mode, for all the tested temperatures. It is of interest to observe that increasing the sensor surface temperature, the response in dark mode increased almost linearly from 1 ppm to 10 ppm of CO (Figure 6a) while under UV irradiation (Figure 6b) the response increase at 150 °C was clearly not linear. The observed behavior was ascribed to the combined effect of the UV photo-activation and the sensor temperature that favored the CO desorption from the sensing surface, as explained in the following discussion (Section 3.3).

The most interesting results were obtained when the sensor surface was irradiated with UV in a non-continuous mode, i.e., “UV On in air” mode. Figure 7 shows the variation of the sensor layer resistance vs. time at the operating temperature of 100 °C, under CO pulses of different concentration. The UV irradiation of the sensor surface in airflow caused a resistance decrease that reverted when the UV LED was switched off and the sensor was exposed to the CO pulse (Figure 7). Increasing CO in air, from 1 ppm to 10 ppm, the observed resistance variation increased showing a correlation with the CO concentration. 

It was clearly noted that, when the UV LED was switched on during the sensor surface regeneration, the sensor resistance in CO, R_CO_, was higher than the resistance in air suggesting a p-type behavior (Figure 7).

In Figure 8a–c are summarized data acquired in all the above described conditions. In each plot of Figure 8 is shown the sensor response to a CO pulse varying the sensor temperature in the three tested UV modes. From data in Figure 8a–c, the room temperature was always a condition of low response for the In_2_O_3_ sensor while for higher temperatures, 100 °C and 150 °C, the CO detection was clearly observed. 

As pointed out before, UV irradiation during CO detection, “UV On in air+CO”, was a condition that did not show a real advantage in terms of sensor response compared to the “UV Off” mode, for all the operating temperatures tested (Figure 8a–c).

Irradiating the sensor surface in the airflow at T = 100 °C and 150 °C, according to curves in Figure 8a–c, the sensor response was <1, like in a p-type semiconductor, where the sensor resistance in CO, R_CO_, resulted higher than the resistance in air, R_0_. For a CO concentration of 10 ppm and T = 100 °C it was (Figure 8a):
$${\left(\frac{R_{0}}{R_{\mathrm{CO}}}\right)}_{\mathrm{UV}\;\mathrm{Off}}=1.2\;\mathrm{and}\;{\left(\frac{R_{0}}{R_{\mathrm{CO}}}\right)}_{\mathrm{UV}\;\mathrm{On}\;\mathrm{in}\;\mathrm{Air}}=\;0.77$$

However, inverting the previous ratio (Figure 8a):
$${\left(\frac{R_{\mathrm{CO}}}{R_{0}}\right)}_{\mathrm{UV}\;\mathrm{On}\;\mathrm{in}\;\mathrm{Air}}=1.3$$

By these results, it appears that the right combination of sensor temperature and UV light operation mode is capable of increasing the sensor performance towards CO detection. In the specific, at the operating temperature of 100 °C, the p-type response of In_2_O_3_ in “UV On in air” mode is higher than in dark or in “UV On in air+CO” for all the tested CO concentrations. The advantage of irradiating the sensor only during the regeneration in airflow, however, was lost when the operating temperature was increased to 150 °C. 

Motivated by the interesting features displayed, we further studied the performances of the sensor in these optimized conditions. The sensitivity of the sensors was then evaluated plotting the sensor responses vs. CO concentration (Figure 9a) for assessing its suitability in the monitoring of CO. The sensor showed good response to the smallest tested concentration of CO (1 ppm).

When the responses are plotted in a log–log scale, a linear trend was observed as a function of the gas target concentration (Figure 9b). The limit of detection, defined as the lower concentration at which the response is significantly differentiated from the noise signal (usually at S/N = 3), was extrapolated to be below 200 ppb. This very low detection limit suggests that the photoactivated In_2_O_3_-based sensor can be applied for the practical use in the field of environmental control.

### 3.3. Sensing Mechanism

Based on results above reported, a discussion on the CO sensing mechanism of the sensor proposed is presented. Indium oxide is a semiconductor with a band gap of 3.5–3.7 eV that shows n-type behavior due to intrinsic defects (oxygen vacancies) [13,22,23]. UV irradiation is generally considered to improve the sensor response because the photons radiation of the semiconductor surface changes the surface potential at the grain boundaries [13,23] enhancing the conductance of the sensing layer [16,17]. In this sense, the UV irradiation can reduce the operating temperature of the metal oxide allowing, in some cases, the use of the resistive sensor at room temperature [15]. In our experiments, however, the continuous UV irradiation of the In_2_O_3_ sensor did not result in an improved response. According to the study of Espid et al. [17], a high UV irradiation of the sensing surface increases the generation rate of electron/hole pairs but also their likelihood of recombination thus favoring the desorption of the molecular and atomic oxygen interacting with the charge carriers UV generated:
(1)$$0\to e_{\left(\mathrm{hv}\right)}^{-}+h_{\left(\mathrm{hv}\right)}^{+}$$
(2)$$O_{2\left(\mathrm{gas}\right)}+e_{\left(\mathrm{hv}\right)}^{-}↔O_{2\left(\mathrm{ads}\right)}^{-}$$
(3)$$O_{2\left(\mathrm{gas}\right)}↔O_{2\left(\mathrm{ads}\right)}^{-}+h_{\left(\mathrm{hv}\right)}^{+}$$

The final effect was that under continuous UV light, the ratio R_0_/R_air+CO_ was similar or even lower than with the UV LED off.

For a similar reason, the UV photo-activation during CO oxidation, “UV On in Air+CO” mode, although caused the lowering of the baseline resistance R_0_, did not improve the sensor response due to the rapid desorption of oxygen even at low temperatures.

The improved sensor response obtained in “UV On in Air” mode is related to the “p-type” behavior shown by the indium oxide layer when irradiated in the airflow. In Figure 10, the baseline resistance at T = 100 °C in UV Off mode, R_0_ = 550 Ω, decreased to R’_0_ = 280 Ω when the UV LED was switched on in the airflow and then increased to R_CO_ = 360 Ω when the UV was off and the sensor exposed to a 10 ppm CO pulse.

The significant decrease of the sensor baseline resistance mainly due to the generation of a large number of electron/hole pairs in the semiconductor particles was followed by an increase in resistance as soon as the UV LED was switched off and the CO + air mixture was sent. 

The charge carriers recombination was the dominant effect in terms of sensor resistance variation prevailing the contemporaneous charges addition due to the CO oxidation:
(4)$$e_{\left(\mathrm{hv}\right)}^{-}+h_{\left(\mathrm{hv}\right)}^{+}\to 0$$
(5)$$\mathrm{CO}+O_{\left(\mathrm{ads}\right)}^{-}\to {\mathrm{CO}}_{2}+e^{-}$$

The combination of the two opposite effects causes that the sensor resistance does not return to the baseline value R_0_ but stops to a value that is proportional to the CO concentration. As shown in Figure 10, the observed p-type behavior was not, in fact, due to a real change in the semiconductor properties of In_2_O_3_ but resulting from the combination of UV illumination and operating conditions.

At the sensor temperature of 150 °C, however, the positive effects of working in the “UV On in air” mode decreased. By increasing the temperature of the MOX semiconductors, different effects are produced, such as the modification of the surface potential of the sensitive layer, the adsorbed oxygen species and their adsorption/desorption rate, the speed of the chemical reactions of the molecules/atoms involved [11,12]. The worsening of the sensor response observed at 150 °C in UV light in regeneration, however, was ascribed to the increase of the recombination rate of the UV-generated charges with temperature [18,24] and the consequent decrease of the ratio R_air+CO_/R_0_.

## 4. Conclusions

The effect of the UV light on the sensing properties of indium oxide in CO detection at low temperatures has been studied by varying the irradiation mode of the sensing layer. Results showed that operating the sensor under UV radiation did not improve the In_2_O_3_ response at low temperatures in comparison to the typical working condition in dark. An improvement was observed when the UV LED irradiated the sensing layer only in airflow, namely in the regeneration phase, and was turned off in detection. The effect, however, was particularly significant at the sensor temperature of 100 °C. In these operating conditions, the enhancement of response observed could be exploited for detecting sub-ppm concentrations of CO (up to 200 ppb) at a relatively low temperature, suggesting that the photoactivated In_2_O_3_ sensor can be applied for the practical use in the field of environmental control.

## Figures and Tables

**Figure 1 micromachines-10-00338-f001:**
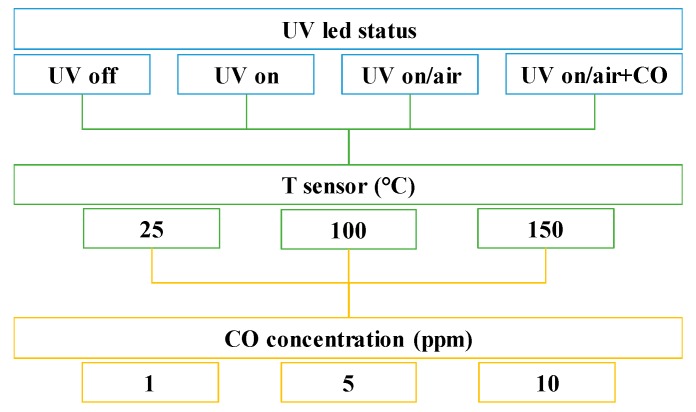
Diagram showing the experimental conditions used.

**Figure 2 micromachines-10-00338-f002:**
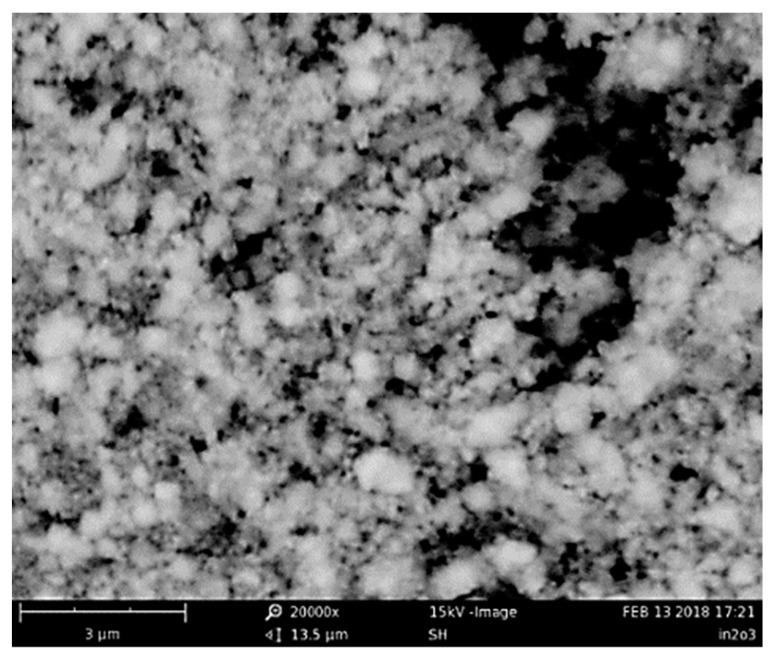
Scanning Electron Microscopy (SEM) image of the prepared In_2_O_3_.

**Figure 3 micromachines-10-00338-f003:**
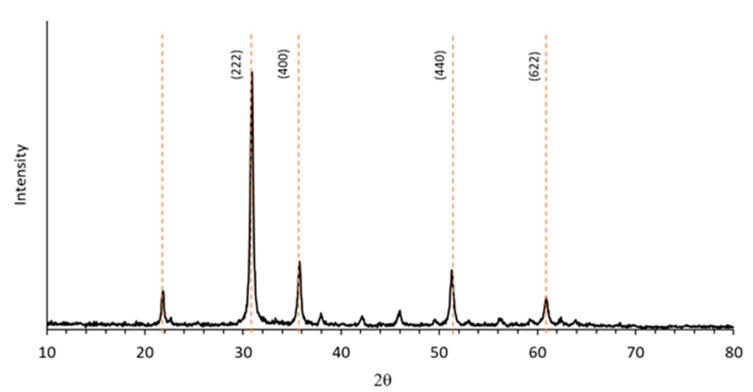
XR diffractogram of the calcined indium oxide powder.

**Figure 4 micromachines-10-00338-f004:**
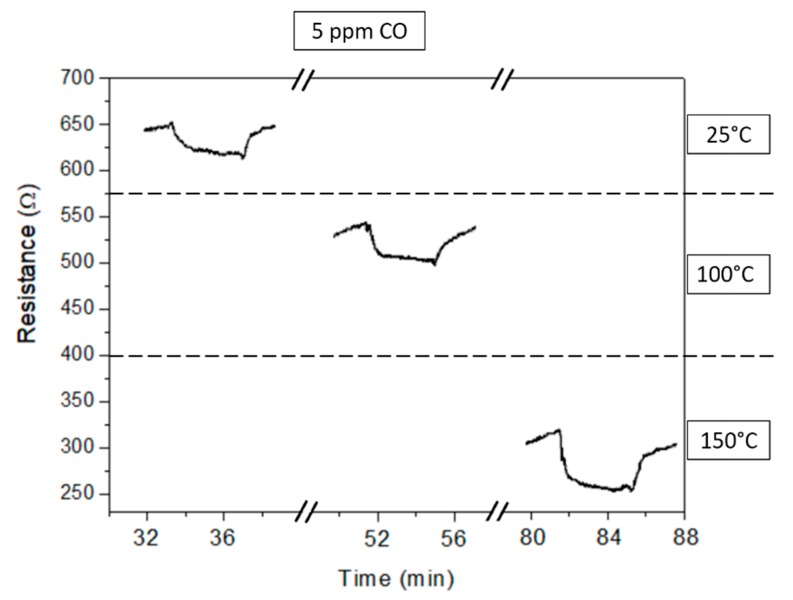
Transient response of the indium oxide sensor to 5 ppm CO at different operating temperatures.

**Figure 5 micromachines-10-00338-f005:**
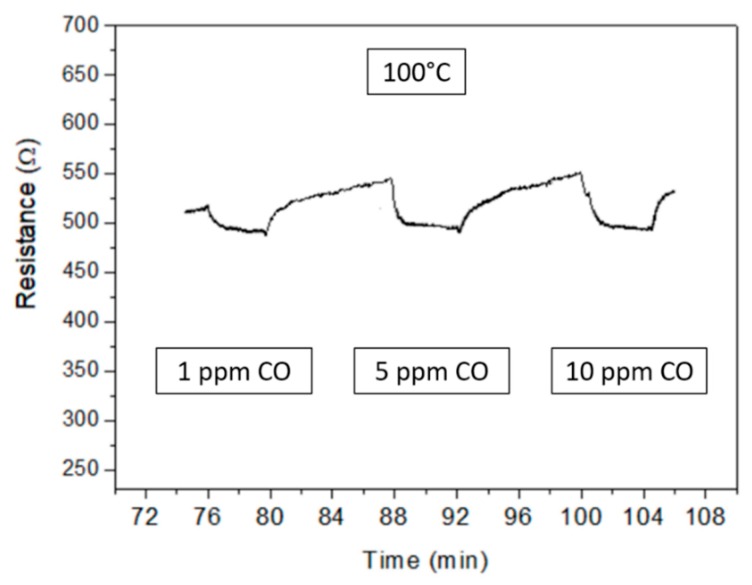
Transient sensor response at T = 100 °C and different CO concentrations.

**Figure 6 micromachines-10-00338-f006:**
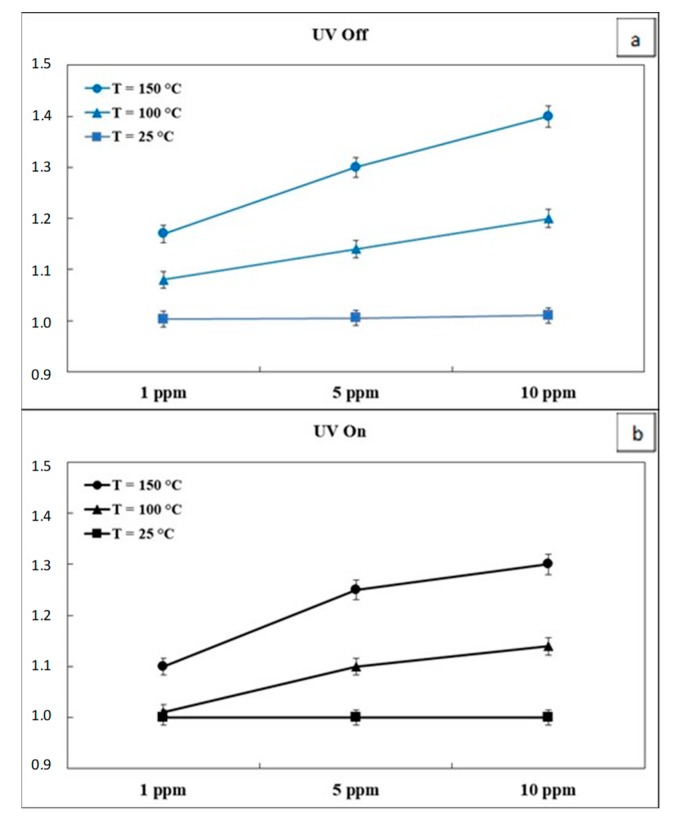
In_2_O_3_ sensor response at different CO concentrations and temperatures in “UV Off” (**a**) and “UV On” (**b**) mode. Error bars are calculated on the average of 5 runs.

**Figure 7 micromachines-10-00338-f007:**
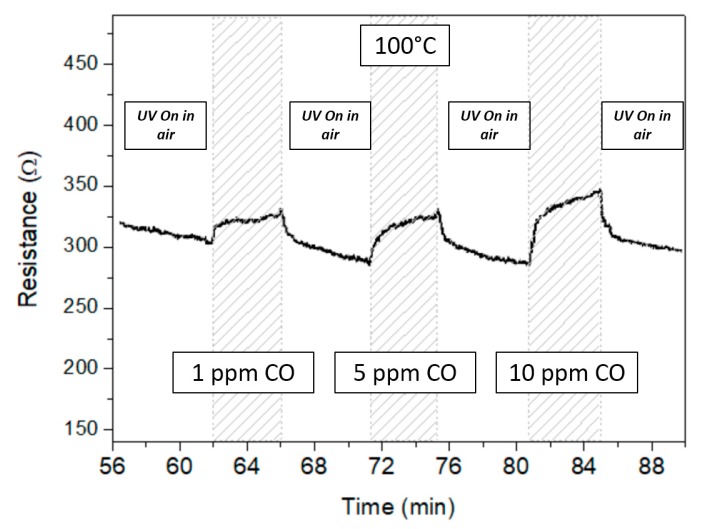
Transient sensor response at T = 100 °C and different CO concentrations in “UV On in air” mode.

**Figure 8 micromachines-10-00338-f008:**
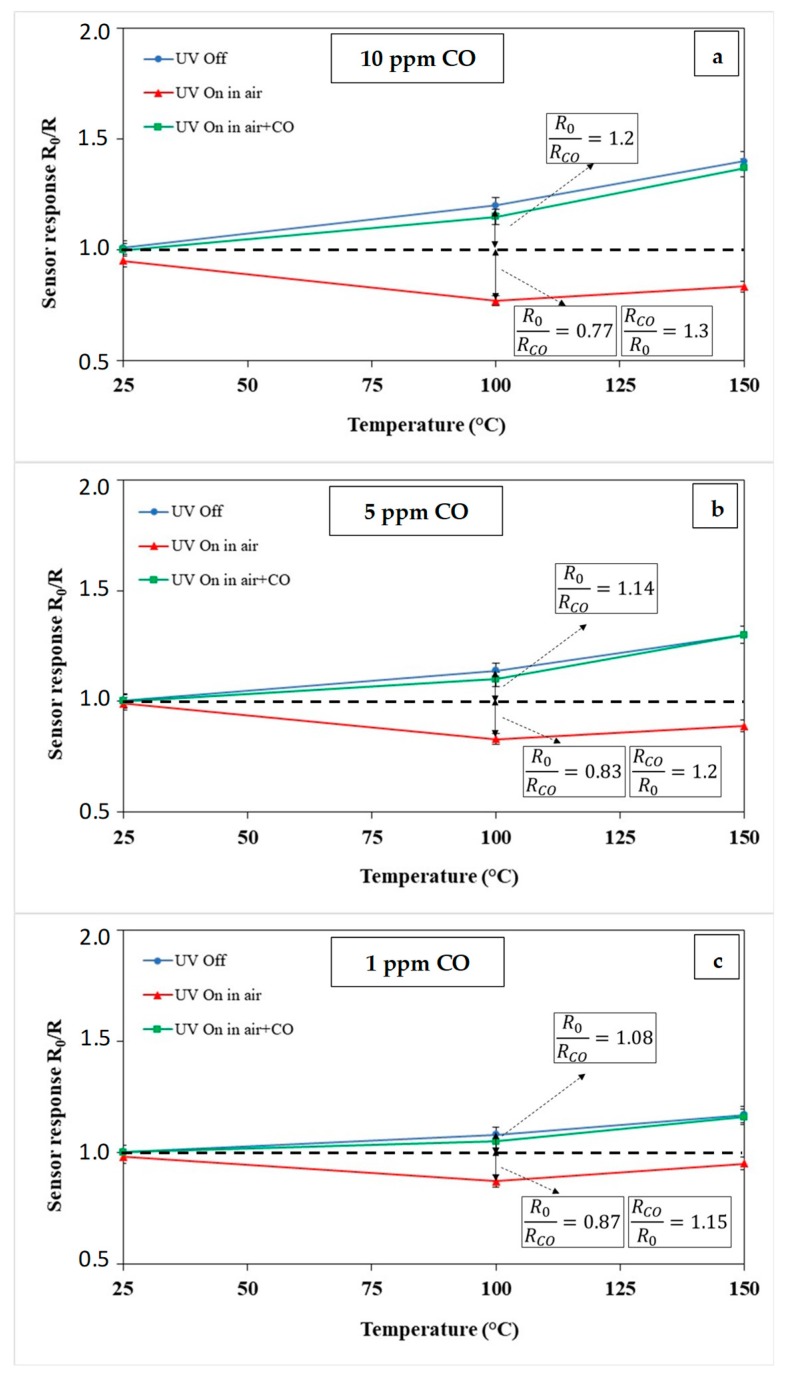
Sensor response to (**a**) 10 ppm; (**b**) 5 ppm and (**c**) 1 ppm of CO at different operating temperatures and different UV irradiation modes. Error bars are calculated on the average of five runs.

**Figure 9 micromachines-10-00338-f009:**
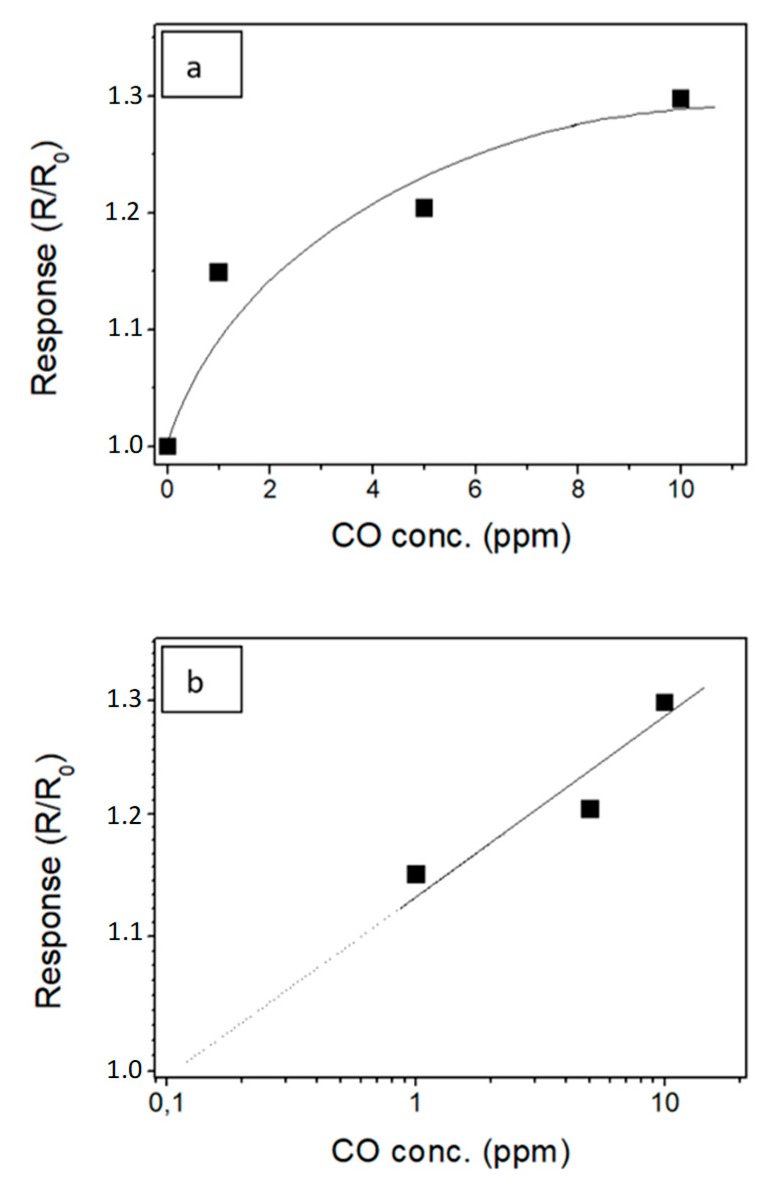
(**a**) Calibration curve showing the sensor response to different CO concentrations under the “UV On in air” method; (**b**) calibration curve plotted in log-log scale.

**Figure 10 micromachines-10-00338-f010:**
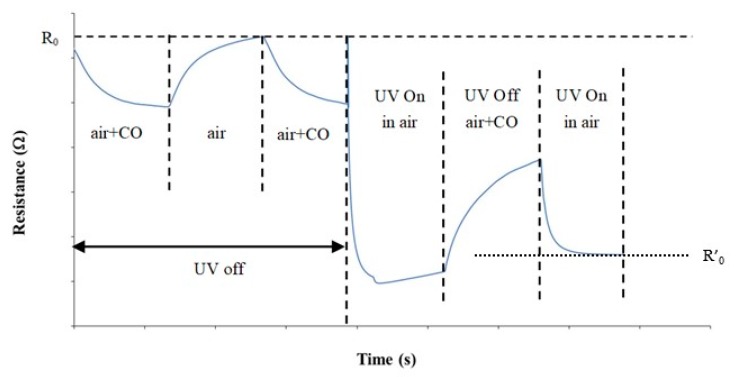
Transient sensor response in “UV Off” mode and “UV On in air” mode. (T = 100 °C, 10 ppm CO).

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
