# Peer review of "Effects of UV Irradiation on the Sensing Properties of In_2_O_3_ for CO Detection at Low Temperature"

_micromachines, 2019, doi:10.3390/mi10050338_

Round 1

Reviewer 1 Report

In this manuscript, the authors studied the effect of UV  in promoting CO sensing on In2O3 -based gas sensors.  Even if the paper relates an original research work, some  important information are missing.  To be published, the manuscript should be improved.    In the Morphological and Microstructural Characterization part,  authors should give some information on the pore size distribution of  the material used.   In the Gas Sensor Measurement part, it is important to have information about the stability over time of the sensor.  And there are no data on the influence of humidity  on the sensing performances.    Additional comments:   -   Please add error bars for figures 6,8 and 9. -   The quality of figures should be improved.

Author Response

Dear Reviewer,

Please find details in attachment.

Reviewer 2 Report

The present paper describes an interesting method to optimize a gas sensor working at low temperatures.  The material and sample preparation procedure is well described and the results are supported by material characterization of the synthetized structures and the sensor characteristic data. But some questions are still open.
The reviewer recommends a major revision. The following points should be considered before publication.

1. All measurements are in dry atmosphere. Did you investigate the sensor behavior in humid atmosphere? How influences humidity the sensor response and behavior? Please comment on this and add, if available, sensor results.

2. The measured resistance is given in ohm. The very low resistance of the metal oxide film, especially at the low temperatures, astonishes me. Why is the resistance such low? Please comment in this.

3. In general, the description and explanation of the presented figures is very short. The figures and presented results should be explained and discussed more in detail, in special Figutres 4 to 8.

4. In Fig. 6, the sensor response S is plotted versus temperature for different CO concentrations. I would prefer a representation of response S versus CO concentration in ppm for the three tested temperatures. From my point of view, this representation would show the results more clearly. Please change the figures 6. and 6b and modify the description.

5. Fig. 7 shows the transient sensor response at 100°C in “UV On in air mode”.  A more detailed description and a discussion of this result is missing and should be added to improve the quality of the paper. The same applies to figure 8. Here, R0/RCO values are visible, but an explanation in the text is missing. Please add some explanation and comment on this.

6. On page 9, the authors wrote: “the photoactivated In2O3-sensor can be applied for practical use in the field of environmental control”.  As I understood, the photoactivated mode corresponds to the mode “UV On in air”. How can be distinguished in practical use (environmental control) whether the UV light must be on or off? Please comment on this.

7. It is discussed that the p-type behavior results from the combination of UV illumination and operating conditions. I am missing a statement or resistance result about the influence of the UV light on the sensor signal in air. How changes the base gas resistance between “dark UV off “ and “UV light on”?  The authors should add a measurement result and should comment on this (with respect to the explanation of Fig. 10) to clarify the contributing effects to p-type behavior.

8. The sections “Author contributions; Funding; Acknowledgements; Conflicts of Interests” are not edited. Please edit them appropriate.

Author Response

(The authors gave the same response as above.)

Round 2

Reviewer 1 Report

The authors have taken into account all the requests for revisions and explanations.

In the sentence "In fact, the BET surface area of the precipitated indium oxide, measured by
nitrogen adsorption method, was 20.3 m2/g and the total Pore Volume = 87 mm3/g. For comparison, the surface area of a commercial In2O3 (Sigma Aldrich) resulted much lower, only 10.2 m2/g whit a pore volume of 42 mm3/g" pleace modify wit by with.

Reviewer 2 Report

The quality of the revised version of the paper was improved. The reviewer comments have been considered by the authors and all points have been discussed in detail. .